# Characterization of Sewage Sludge and Food Waste-Based Biochar for Co-Firing in a Coal-Fired Power Plant: A Case Study in Korea

**Yoonah Jeong, Ye-Eun Lee and I-Tae Kim ***

Environmental Resource Research Center, Department of Land, Water and Environment Research,
Korea Institute of Civil Engineering and Building Technology, Goyang 10223, Korea;
yoonahjeong@kict.re.kr (Y.J.); yeeunlee@kict.re.kr (Y.-E.L.)
* Correspondence: itkim@kict.re.kr; Tel.: +82-31-995-0885

**Abstract:** Biomass co-firing in coal-fired power plants has been widely accepted to reduce the environmental burden. In this study, food waste (FW) and sewage sludge (SS), which are the main types of municipal organic waste, were selected as solid refuse fuel (SRF). To compensate for the limitations of FW and SS, a mixture of FW and SS with varying ratios was processed using pyrolysis and desalination. The fuel properties such as the calorific value, chlorine content, alkali and alkaline earth metallic species (AAEMs) content, and heavy metal content were determined. The calorific values of all biochars were greater than 12.6 MJ/kg, which satisfies the national threshold of Bio-SRF in Korea. Chlorine and AAEMs contents exhibited clear trends for the FW ratio and pyrolysis temperature. Increasing concentrations of heavy metals were observed with increasing SS ratio and pyrolysis temperature. These results provide important insights into the practical application of municipal waste-based biochar in coal-fired plants, as well as the influence of mixing ratio and pyrolysis temperature.

**Keywords:** Bio-SRF; sewage sludge; food waste; pyrolysis; biochar; co-firing

## 1. Introduction

Thermoelectric coal-burning plants account for 38% of global primary energy consumption [1]. Coal combustion is a major source of energy; however, it causes air pollution with the emission of significant amounts of $CO_2$. According to the International Energy Association (IEA), coal is responsible for 9.8 Gt of $CO_2$ emissions [2]. To alleviate air pollution from thermoelectric plants, the co-combustion of biomass and coal has been proposed [3,4].

The potential of biomass as a solid recovered fuel (SRF) has been evaluated in previous studies [5–7]. The application of SRF in thermoelectric plants helps curtail reliance on coal and its negative environmental impacts. It also promotes the efficient management of organic waste as it does not require additional energy for treatment and can be used as a renewable energy resource [3,5]. Various types of biomass are available as co-firing materials, including agricultural residue, algae, wood-processing residue, sewage sludge, and food waste. Furthermore, the application of biochar with coal reduces greenhouse gas emissions because biomass is considered to be carbon neutral. However, the inherent characteristics of biomass should be considered for its practical application. Biomass has a low calorific value, high volatile content, and high moisture content, which may reduce thermal efficiency during plant operation [7]. In addition, high chlorine, alkali, and alkali earth metal (AAEM) contents may lead to slagging and fouling issues [8].

Among the various types of biomass, this study focuses on municipal organic waste, i.e., food waste (FW) and sewage sludge (SS). FW and SS have previously been identified as sources of renewable

energy in the form of biogas, bio-oil, and biochar [9–13]. The simultaneous treatment of FW and SS has been conducted in several ways: anaerobic co-digestion of FW and SS [14], dark fermentation [15], and co-pyrolysis [13]. According to Huang et al. [12], the co-pyrolysis of SS, rice straw, and sawdust produces biochar with a higher content of organic matter and lower yield of biochar.

Increasing attempts to use municipal waste with coal in thermal electric plants have revealed some limitations. First, SS contains a large amount of heavy metals, which will presumably contribute to air pollution [16]. Various heavy metals have been detected in SS, with variable profiles that differ with sampling time and site. Second, the high chlorine content in FW requires a desalination procedure before co-firing as the presence of chlorine leads to the formation of dioxins and dioxin-like compounds as byproducts, which can cause a serious health risk and the corrosion of plant facilities [17,18]. Third, AAEMs in biomass impact ash-related issues such as slagging and fouling [19,20]. As FW typically contains a substantial amount of chlorine and AAEMs, water or acid washing pretreatment procedures may be adopted for de-ashing and desalination. Finally, sufficient biomass calories should be guaranteed to maintain the efficiency of electrical power generated in a coal-fired plant. The lower calorific value of biomass implies that a greater amount of biomass must be combusted to generate the same electrical power. Therefore, to compensate for the abovementioned limitations of FW and SS, this study analyzes different mixtures of FW and SS for application as Bio-SRF in a coal-fired power plant. It is hypothesized that Bio-SRF based on an appropriate proportion of FW and SS will exhibit lower concentrations of heavy metals, chlorine, and AAEMs, as well as a satisfactory calorific value.

Therefore, this study aims to characterize biochar based on food waste and sewage sludge to derive the optimal co-firing material for thermoelectric plants. Bio-SRF is prepared with varying ratios of FW and SS and treated by pyrolysis at 300–500 °C and following desalination. The specific objectives are (1) to determine the calorific value of Bio-SRF and (2) to explore heavy metal and element contents and their relationship to the FW to SS ratio. The specifications of the resulting Bio-SRF are compared with the regulatory criteria of the Republic of Korea [21] and the international standard BS EN 15359:2011 [22].

## 2. Materials and Methods

### 2.1. Materials and Production of Biochar

SS was collected from the Ilsan wastewater treatment plant in the Republic of Korea. SS was prepared by dewatering and mixing sludges from primary and secondary sedimentation basins. Artificial FW was prepared using grains (16%), vegetables (51%), fruits (14%), meat (4%), fish (12%), and eggshell (3%) [23]. The composition and specific ratio of FW followed the protocol proposed by the Ministry of Environment, "Standard Food Waste Sample" [24]. In Korea, Food waste is separately discharged and transported into the treatment facility, not collected together with other municipal wastes. As a large quantity of food waste is combined and treated in a mass, the heterogeneous nature of food waste from various origin can be offset and the averaged composition of food waste can be estimated. Basic information, including the major elemental composition, can be found in a previous study [23]. The compositions and atomic ratios of FW were as follows: C (51.5%), H (13.2%), N (3.1%), O (32.2%), C/N (16.38), H/C (0.26), and O/C (0.63). The mixture ratios of SS and FW were set to 100:0, 66.7:33.3, 50:50, 33.3:66.7, and 0:100, and referred to as SS100, FW33, FW50, FW67, and FW100, respectively. Artificial FW samples and SS were dried at 105 °C for 24 h, then mixed and pulverized to ensure homogeneity of samples.

The experimental set-up, including pyrolysis and desalination, was the same as that of a previous study [25]. Briefly, the pyrolysis of a mixture of SS and FW in various ratios was performed using a customized pyrolyzer (Handuk R-FECO Co., Ltd., Gyeonggi-do, Republic of Korea). The pyrolyzer consists of an electric furnace with sampling chamber, cooler, and gas combustor. Nitrogen gas was supplied at 5 L/min during the operation and biogas from pyrolysis unit was combusted. The size of the sample tray was $50 \times 200 \times 45$ mm. Pyrolysis temperatures of 300 °C, 400 °C, and 500 °C were tested

and the heating rate was 10–15 °C/min. At each pyrolysis temperature, the retention time was 60 min. After pyrolysis, the samples were washed with water for desalination (1:10 *w/v* ratio). Desalination was performed for 30 min and washed biochar was filtered with a glass fiber filter (0.7 um, Whatman®). The filtered biochar was dried overnight at 105 °C. The samples were then stored in a centrifuge tube until further analysis.

### 2.2. Analysis

All Bio-SRF samples were characterized with respect to their calorific value, chlorine content, and elemental composition, as well as by proximate analysis including moisture, volatile, fixed carbon, ash, and heavy metal contents. A bomb calorimeter (6300 Calorimeter A1200 DDEE, Parr, IL, USA) was used to determine the calorific value based on ASTM D 5468-95. Elemental and metal analyses were conducted using an inductively coupled plasma atomic emission spectrometer (ICP-AES, ICP-730ES, Varian, Australia) and inductively coupled plasma mass spectrometry (ICP-MS, Varian 820-MS, Varian, Australia) after microwave digestion. Proximate analysis was performed using KS E 3804. For sample preparation, a microwave sample digestion system (MARS-6, CEM Corp, USA) based on EPA 3051A was applied. Chlorine concentration was measured before desalination and AAEM and heavy metal contents of biochar were determined after desalination. All data work-up was processed using Origin 2019b (OriginLab Corporation, Northampton, MA, USA).

## 3. Results and Discussion

### 3.1. Calorific Value

The calorific values of all biochar samples are presented in Figure 1. The calorific value gradually increases with increasing FW ratio and ranges from 12.6 to 27.1 MJ/kg, 15.5 to 28.6 MJ/kg, and 18.0 to 27.8 MJ/kg at pyrolysis temperatures of 300 °C, 400 °C, and 500 °C, respectively. Three apparent trends are shown in Figure 1. First, the addition of FW in biochar clearly augments the fuel efficiency, which results in a 115.1% increase in the calorific value at a pyrolysis temperature of 500 °C. The calorific value depends on the relative proportions of carbon, hydrogen, and oxygen in the biochar. In general, the C content in FW is greater than that in SS, leading to a higher calorific value, as shown in Figure 1. This implies that a greater amount of SS100 than FW100 would be required to generate the same electric power because of the low calorie of SS100. Considering that the calorific value is the most critical factor of the fuel, SRF with a higher proportion of FW is desirable for the practical operation of a coal-fired plant.

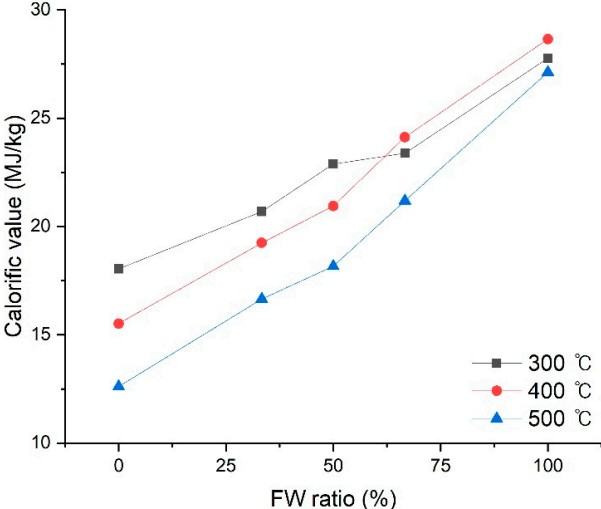

**Figure 1.** Calorific value (MJ/kg) of SS100, FW33, FW50, FW67, and FW100 under pyrolytic temperatures of 300 °C, 400 °C, and 500 °C.

Second, a low calorific value is observed at higher pyrolysis temperatures. This result indicates that increasing the pyrolytic temperature reduces the energy content of biochar. Lastly, the pyrolytic temperature also affects the rate of calorific value increase. The slopes presented in Figure 1 increase from 22.8 (300 °C) to 31.7 (400 °C), then 34.4 (500 °C). In turn, the difference of calorific values in SS100 samples range from 12.6–18.0 MJ/kg, which is an increase of 43.1%. This difference decreases to 5.6% (27.1–28.6 MJ/kg) for FW100. A possible explanation for this is the influence of pyrolysis temperature on volatile matter and carbon content [26]. At higher pyrolysis temperatures, the final product of biochar may contain less volatile matter and concentrated carbon than the raw mixture of FW and SS, leading to a higher heating value [27,28]. Consistent with the findings of previous studies, this study reveals decreased volatile contents and increased fixed carbon at higher pyrolysis temperatures (Table 1). Additionally, the volatile content does not change significantly with the FW ratio, whereas the fixed carbon content increases with an increasing FW ratio, implying that condensed carbon predominantly originates from FW. As the organic component of SS is easily decomposed, most of the carbon content in SS is volatilized during pyrolysis [29]. Such characteristics of SS lead to a lower calorific value at higher pyrolysis temperatures.

**Table 1.** Proximate analysis of a raw mixture of food waste (FW) and sewage sludge (SS) (dried at 105 °C) and biochar pyrolyzed at 300 °C, 400 °C, and 500 °C. Fuel ratio (unitless) is calculated by dividing fixed carbon content by volatile content.

| Sample Description | FW:SS Ratio | Moisture (%) | Volatile (%) | Ash (%) | Fixed Carbon (%) | Fuel Ratio (Unitless) |
|---|---|---|---|---|---|---|
| Raw mixture of FW and SS | 100:0 (FW100) | 7.07 | 77.20 | 3.09 | 19.70 | 0.26 |
| | 67:33 (FW67) | 12.20 | 72.50 | 10.60 | 16.90 | 0.23 |
| | 50:50 (FW50) | 10.40 | 72.00 | 13.10 | 15.00 | 0.21 |
| | 33:67 (FW33) | 12.00 | 67.70 | 17.70 | 14.60 | 0.22 |
| | 0:100 (SS100) | 11.40 | 63.80 | 23.70 | 12.50 | 0.20 |
| Pyrolysis at 300 °C | 100:0 (FW100) | 1.71 | 61.30 | 4.33 | 34.40 | 0.56 |
| | 67:33 (FW67) | 0.46 | 59.00 | 14.10 | 26.90 | 0.46 |
| | 50:50 (FW50) | 1.28 | 58.90 | 16.40 | 24.70 | 0.42 |
| | 33:67 (FW33) | 1.80 | 54.30 | 23.30 | 22.40 | 0.41 |
| | 0:100 (SS100) | 3.34 | 55.20 | 29.20 | 15.60 | 0.28 |
| Pyrolysis at 400 °C | 100:0 (FW100) | 1.34 | 32.30 | 7.20 | 60.50 | 1.87 |
| | 67:33 (FW67) | 1.19 | 37.30 | 20.30 | 42.40 | 1.14 |
| | 50:50 (FW50) | 0.68 | 30.90 | 28.50 | 40.60 | 1.31 |
| | 33:67 (FW33) | 2.76 | 32.20 | 34.00 | 33.80 | 1.05 |
| | 0:100 (SS100) | 2.16 | 30.50 | 42.80 | 26.60 | 0.87 |
| Pyrolysis at 500 °C | 100:0 (FW100) | 3.40 | 19.50 | 9.24 | 71.20 | 3.65 |
| | 67:33 (FW67) | 2.67 | 21.60 | 27.60 | 50.80 | 2.35 |
| | 50:50 (FW50) | 1.10 | 19.30 | 38.20 | 42.50 | 2.20 |
| | 33:67 (FW33) | 3.79 | 17.70 | 40.00 | 42.30 | 2.39 |
| | 0:100 (SS100) | 5.90 | 19.50 | 50.50 | 30.00 | 1.54 |

The observed calorific values are compared with two regulatory thresholds: Bio-SRF requirements of the Republic of Korea, "Enforcement rule of the act on the promotion of saving and recycling of resources" (Enforcement date 27 May 2020) (Ordinance of Prime Minister No. 869, 27 May 2020, Partial Amendment), and BS EN 15359:2011 (Solid recovered fuels—Specifications and classes) [21,22]. The minimum calorific value required for Bio-SRF is 3000 kcal/kg (i.e., 12.6 MJ/kg) (Table 2). BS EN 15359:2011 classifies fuel by three criteria: calorific value, chlorine content, and mercury content. First-class fuel under BS EN 15359:2011 requires over 25 MJ/kg, whereas second-class fuel requires over 20 MJ/kg. Thus, most samples in this study satisfy the calorific threshold of Bio-SRF. Biochar with a high proportion of FW has a higher calorific value and all FW 100 biochar samples fulfill the first-class criteria for BS EN 15359:2011.

**Table 2.** Specifications of biomass-solid refuse fuel (Bio-SRF) based on the "Enforcement rule of the act on the promotion of saving and recycling of resources" (Enforcement date 27 May 2020) (Ordinance of the Prime Minister No. 869, 27 May 2020, Partial Amendment) in Republic of Korea [21].

| Characteristic | | Unit | Pellet | | Non-Pellet | |
|---|---|---|---|---|---|---|
| Shape and size | | mm | Diameter | ≤50 | Width | ≤120 |
| | | | Length | ≤100 | Length | ≤120 |
| Moisture | | wt.% | ≤10 | | ≤25 | |
| Net calorific value | | kcal/kg | Imported SRF ≥ 3150 Manufactured SRF ≥ 3000 | | | |
| Ash | | wt.% | ≤15 | | | |
| Chlorine | | wt.% | ≤0.5 | | | |
| Sulfur | | wt.% | ≤0.6 | | | |
| Biomass | | wt.% | ≥95 | | | |
| Metal | Hg | mg/kg | ≤0.6 | | | |
| | Cd | | ≤5.0 | | | |
| | Pb | | ≤100 | | | |
| | As | | ≤5.0 | | | |
| | Cr | | ≤70.0 | | | |

### 3.2. Chlorine Content

Incineration of waste has been identified as a major source of polychlorinated dibenzo-p-dioxin (PCDD) and dibenzofuran (PCDF) emissions [18]. As the presence of chlorine triggers the formation of PCDD/Fs, low-chlorine-content biomass is a regulatory requirement. According to the regulatory threshold (Table 2), the chlorine content should be less than 0.5%. Conversely, first-class SRF under BS EN 15359:2011 should have a chlorine content of less than 0.2%.

In this study, chlorine concentrations are 0.18–1.06%, 0.22–1.24%, and 0.17–1.77% at pyrolysis temperatures of 300 °C, 400 °C, and 500 °C, respectively. Figure 2 reveals a gradual increase in chlorine content with an increasing FW ratio. As a high concentration of chlorine is contained in FW, an increase in the chlorine concentration of biochar with an increasing FW ratio is expected. All SS100 samples exhibit a chlorine content of less than 0.5%, which adheres to Bio-SRF criteria. Except for that produced under a pyrolysis temperature of 400 °C, the SS100-based biochar samples satisfy BS EN 15359:2011. Note that chlorine concentration in Figure 2 was measured before the desalination procedure. Therefore, less chlorine content is expected to retain in the final product of SRF after water washing. A sudden drop is observed in the chlorine concentration in FW50 at pyrolysis temperatures of 400 °C and 500 °C, as shown in Figure 2. This could be attributed to the moisture content of biochar (Table 1), which exhibits a similar pattern to chlorine. Chlorine is present in biochar as alkali metal chlorides such as NaCl and KCl; therefore, the chlorine content can be related to the moisture content. It is difficult to generalize the unexpected drop in chlorine and moisture content observed in this study, which remains speculation.

Approximately half of the chlorine content is released from biochar during pyrolysis at temperatures below 500 °C, mainly in the form of HCl [30]. At pyrolytic temperatures above 800 °C, complete removal of chlorine is expected during pyrolysis [31]. In this study, a high chlorine concentration remains in the biochar, probably because the low pyrolysis temperature limits the complete volatilization of chlorine during pyrolysis. Accordingly, a water washing procedure should be conducted for desalination to satisfy the regulatory criteria for chlorine concentration. To efficiently remove the chlorine in biochar, either a high pyrolysis temperature of greater than 800 °C or desalination such as water washing or acid treatment can be viable options for lowering the chlorine content.

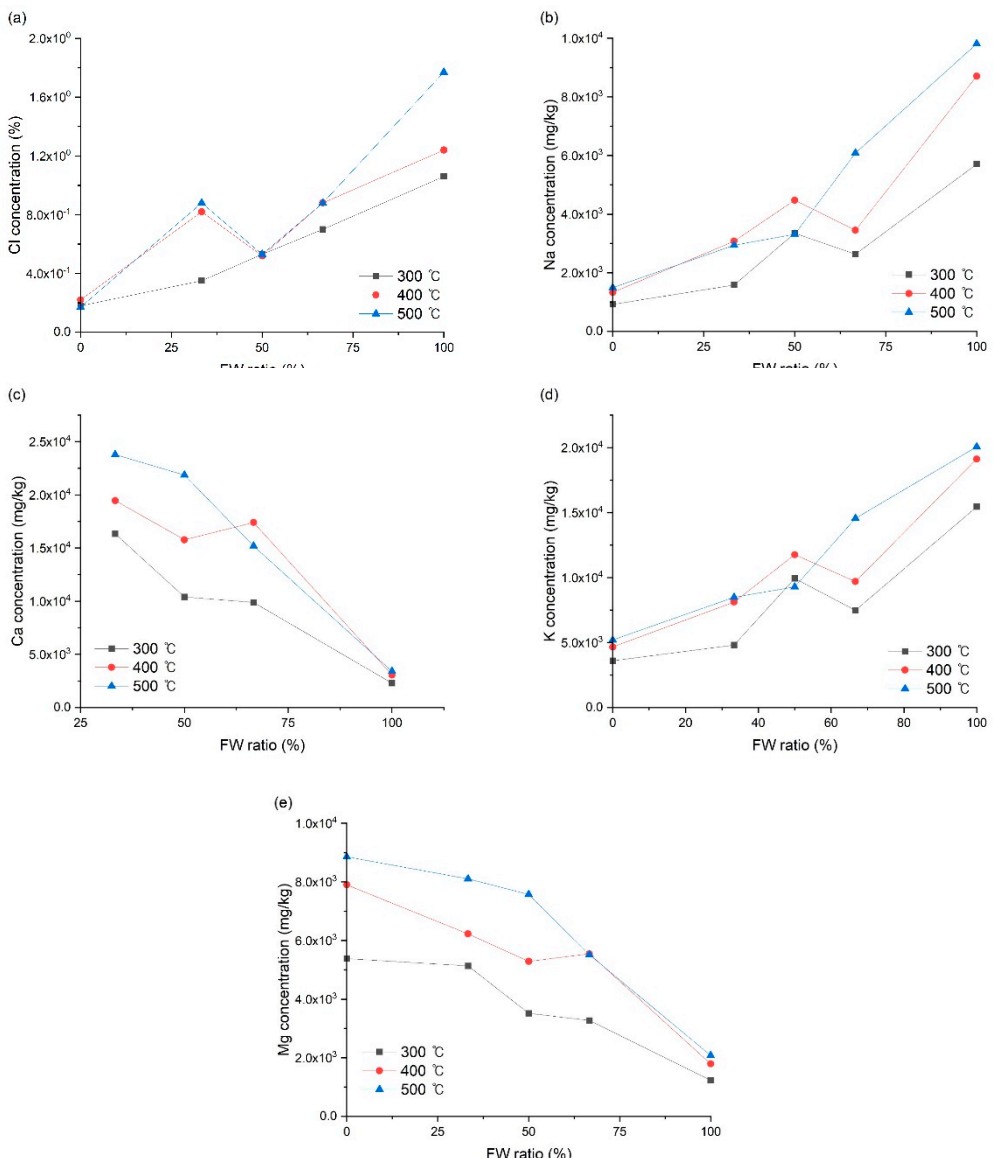

**Figure 2.** Elemental composition of biochar samples produced by various mixing ratios of FW and SS: (**a**) chlorine content and alkali, and alkali earth metals AAEMs contents (**b**) Na, (**c**) Ca, (**d**) K, (**e**) Mg. Elemental concentrations were measured after pyrolysis at 300–500 °C and desalination.

### 3.3. AAEMs Content

Combustion of biomass, including coal, SS, and FW, generates solid waste in the form of ash. Ash-related issues are linked to the amount and composition of the inherent AAEMs. It has previously been observed that the co-firing of biomass with coal increases the liquid ratio in fly ash due to the high AAEMs content of biomass [32]. To estimate the synergistic effect of municipal waste and coal, the AAEMs contents of biochar with varying ratios of FW and SS were investigated. Furthermore, the molar distribution of SRF was calculated using the following equation [8]:

$$\text{Molar distribution} = (2\text{Ca} + 2\text{Mg} + \text{Na} + \text{K})/(2\text{S} + \text{Cl})$$

A comparison of the molar distribution to one implies the relative distribution of AAEMs, sulfate, and chloride in SRF. The calculated molar distribution ranges from 1.91 to 4.66 (Figure 3), implying that approximately 20–50% of AAEMs are present in the form of sulfate and chloride. At low pyrolytic temperatures (i.e., 300 °C), the ratio of FW in biochar does not significantly influence the molar

distribution; however, the influence of pyrolytic temperature becomes clear at 400 °C and 500 °C. This is probably because alkali salts generally start to be released from biochar above 300 °C [26]. The form of sulfate and chloride with AAEMs in biochar decreases with increasing SS ratio and pyrolytic temperature.

Regarding the individual AAEM components, Na and K concentrations increase with increasing FW ratio, whereas Ca and Mg concentrations decrease (Figure 2b–d). Overall, biochar samples prepared with FW and SS are enriched with AAEMs components. The influence of pyrolytic temperature is clear for all AAEMs components. Higher concentrations are observed at higher temperatures because of concentrated AAEMs with pyrolytic temperatures. A significant proportion of AAEMs is expected to volatilize during pyrolysis. Rearranging and restructuring of the chemical bonds in biochar is generally initiated at 350 °C [33]. Specifically, the majority of Na and K, which are monovalent cations, are vaporized and released from biochar in the form of aerosols, whereas a certain amount of Ca and Mg, which are divalent cations, are retained in the biochar [34]. This is because the volatilization of calcium during pyrolysis occurs at higher pyrolytic temperatures than that of Na and K [31,35]. In this study, the temperature was set to 300–500 °C, indicating that a certain amount of AAEMs is retained in the biochar.

It should be noted that the removal of AAEMs from biomass is not mandatory for the co-firing of biomass because (1) there are no regulatory guidelines for AAEMs concentration in biomass and (2) the content of AAEM species does not significantly affect the yields of biochar or bio-oil [36]. However, reducing AAEM species is desirable because the content and composition of AAEMs affect the properties of pyrolyzed products and byproducts (e.g., ash and syngas). To reduce the AAEMs concentration in biochar, a higher pyrolytic temperature or fortified washing treatment can be applied. When the pyrolysis temperature is sufficient (i.e., over 1000 °C), most of the AAEMs are evaporated and released from the biochar. However, as the application of higher pyrolysis temperatures consumes a lot of electric power, biochar washing after pyrolysis can be a practical option. In this study, the water washing approach was efficient for removing chlorine, but not AAEMs. Therefore, $CO_2$-saturated water and/or acid treatment should be applied in further studies [36].

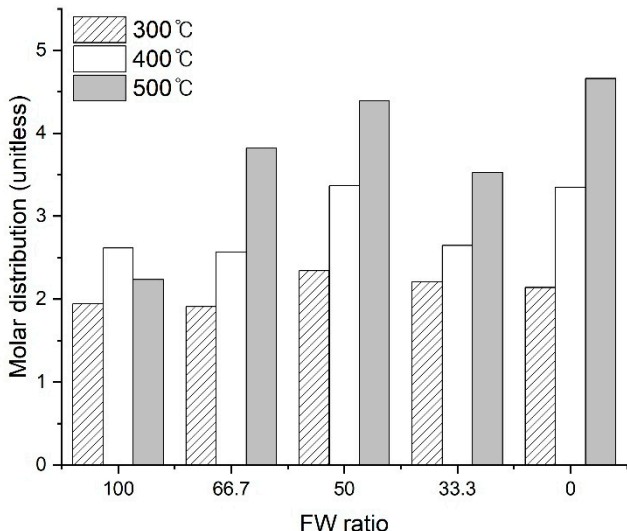

**Figure 3.** Molar distribution (unitless) of biochar samples under various pyrolytic temperatures (300 °C, 400 °C, and 500 °C).

### 3.4. Heavy Metal Content

SS is a byproduct of wastewater treatment plants (WWTPs) that contains not only high amounts of organic matter but also various heavy metals. Accordingly, a high heavy metal concentration is expected in SRF with a higher SS ratio. The contents of seven heavy metals were monitored: arsenic (As),

cadmium (Cd), chromium (Cr), copper (Cu), mercury (Hg), lead (Pb), and zinc (Zn). These values were compared with the regulatory concentration standards stated in the Bio-SRF guidelines (Table 2).

Figure 4 presents the results of the heavy metal analysis. Among the target metals, mercury is not detected in any sample. The concentrations of As and Pb at 400 °C, Cr at 300 °C, and As at 500 °C are not presented in Figure 4 due to being lower than the detection limits or not present. Overall, a higher metal concentration is observed at a higher sludge ratio, as expected [37,38]. For all heavy metals, the highest concentration is found in SS100 at a pyrolysis temperature of 500 °C: Zn 1437.4 mg/kg, Cu 738.5 mg/kg, Cr 37.6 mg/kg, Pb 22.1 mg/kg, As 2.8 mg/kg, and Cd 1.9 mg/kg. In contrast, the lowest metal concentrations are found at FW100. Except for copper and zinc, the other four metals exhibit concentrations of less than 1 mg/kg. Zinc and copper account for over 97% of the total heavy metal content, wherein the concentration of zinc is approximately twice that of copper. This significant proportion of zinc and copper in SS-based biochar agrees with the results of previous studies [39].

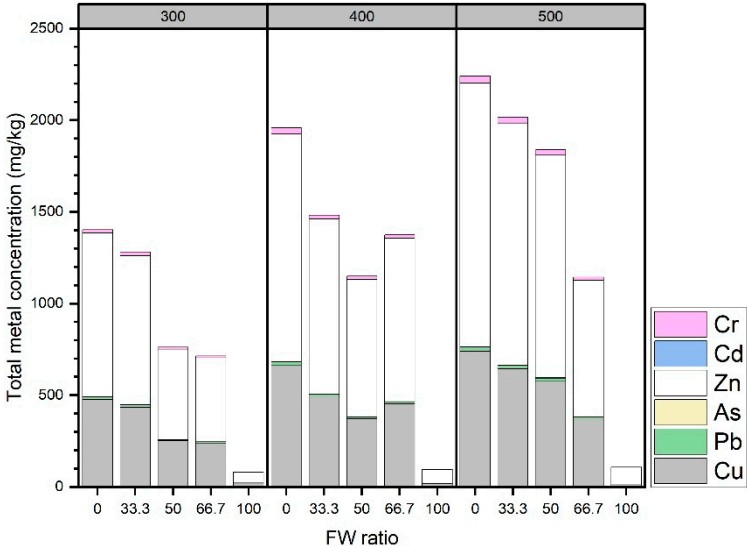

**Figure 4.** Total concentrations of six heavy metals (mg/kg) with varying FW ratios. Metal concentrations of biochar pyrolyzed under temperatures of 300 °C, 400 °C, and 500 °C are presented as cumulative values.

Furthermore, the total heavy metal concentration gradually increases with pyrolysis temperature (Figure 4). This is because a higher pyrolysis temperature enhances the stability of metals in biochar [39]. Among the heavy metals, Cd can be released at a pyrolysis temperature of over 600 °C due to its volatile nature [40]; however, other metals appear to become concentrated with pyrolysis temperature. The increase in heavy metal concentration with increasing sludge ratio is not always linear (Table 3). At a pyrolysis temperature of 400 °C, the zinc concentration decreases from 890.0 mg/kg (FW67) to 749.1 mg/kg (FW50). This characteristic is also observed in Cu (452.1 mg/kg to 371.4 mg/kg), Cr (18.0 mg/kg to 17.2 mg/kg), and Pb (11.5 mg/kg to 9.5 mg/kg). However, this sudden drop is not observed at a pyrolysis temperature of 500 °C. As the sample amounts are not sufficient for multiple measurements, it is difficult to generalize this temporary decrease in heavy metal concentration; thus, further study is required.

The concentrations of heavy metals in most biochar samples are greater than the national guidelines of Korea (Table 2). In particular, the considerable amounts of copper and zinc exceed the regulatory criteria of 10 mg/kg and 100 mg/kg, respectively. Heavy metals are not completely volatilized during pyrolysis or removed during desalination, but retained in the biomass. The only exception is FW100, whose metal concentrations are either negligible or lower than the detection limits. Heavy metals are difficult to remove from biochar because of their stable and immobile nature [41], in which they bind to the organic matter in biochar and sulfides. In addition, the pyrolytic temperature applied in this study is not sufficient to break the metal-biochar bond and release the metal components. The results

of this study indicate that a higher proportion of FW in biochar is advantageous with acceptable heavy metal concentrations.

**Table 3.** Concentrations of six heavy metals (Cu, Pb, As, Zn, Cd, and Cr) (mg/kg) under pyrolysis temperatures of 300 °C, 400 °C, and 500 °C.

| Pyrolysis Temperature | FW Ratio | Cu | Pb | As | Zn | Cd | Cr |
|---|---|---|---|---|---|---|---|
| 300 °C | FW100 | 19.28 | 0.63 | 0.25 | 59.67 | 0.06 | N.D. |
| | FW67 | 238.76 | 5.51 | 1.57 | 457.99 | 0.55 | 10.00 |
| | FW50 | 251.02 | 6.39 | 1.43 | 491.27 | 0.61 | 10.55 |
| | FW33 | 434.03 | 11.24 | 2.28 | 813.10 | 1.03 | 18.95 |
| | SS100 | 475.92 | 12.58 | 2.76 | 892.16 | 1.10 | 18.32 |
| 400 °C | FW100 | 16.96 | N.D. | N.D. | 78.05 | 0.09 | 0.23 |
| | FW67 | 452.05 | 11.48 | 1.28 | 889.99 | 1.05 | 18.04 |
| | FW50 | 371.39 | 9.51 | 1.55 | 749.11 | 0.87 | 17.16 |
| | FW33 | 493.50 | 12.57 | 1.81 | 951.04 | 1.12 | 21.75 |
| | SS100 | 661.05 | 18.81 | 3.05 | 1242.89 | 1.62 | 31.19 |
| 500 °C | FW100 | 10.51 | 0.43 | N.D. | 96.27 | 0.07 | 0.22 |
| | FW67 | 369.00 | 10.07 | 1.31 | 744.99 | 0.86 | 17.21 |
| | FW50 | 575.32 | 17.39 | 1.85 | 1216.10 | 1.48 | 25.40 |
| | FW33 | 642.61 | 20.00 | 2.27 | 1318.09 | 1.77 | 29.67 |
| | SS100 | 738.49 | 22.08 | 2.83 | 1437.43 | 1.88 | 37.55 |

N.D. stands for "not detected.".

*3.5. Application of FW- and SS-Based Biochar as Bio-SRF*

Previous and existing treatment approaches for municipal waste include incineration, landfilling, and mechanical biological treatment (e.g., composting and anaerobic digestion). Landfill and incineration lead to serious environmental pollution and consume a lot of energy, whereas mechanical biological treatment is limited by low efficiency and burdensome management. Considering that the amount of FW and SS is increasing, a realistic and efficient management strategy is urgently required.

In Korea, the renewable energy portfolio standard (RPS) was introduced in 2012 to endorse the development of renewable energy. With the enforcement of the "Act on the Promotion of the Development, Use, and Diffusion of New and Renewable Energy (Act No. 17169, 1 July 2020)" entities engaged in electricity generation of over 500 MW have an obligation to use new and renewable energy. The total amount of power generated by new and renewable energy is determined by a presidential decree on a yearly basis within 10%. Under this regulatory guideline, the SRF to coal ratio was set at 5% in 2019, and an increased ratio of SRF of up to 10% is planned for after 2023.

As the loading ratio of SRF to coal is typically less than 10% [42], the fuel characteristics determined in this study should be interpreted with caution. Note that both SS and FW have heterogeneous characteristics, which could result in inconsistent fuel properties including calorific value, chlorine content, and heavy metal content. Although various types of biomass have been applied for co-firing with coal [43], it is still difficult to standardize the preparation of municipal solid waste-based biochar and predict the influence of SRF in coal-fired plants. The similarity of fuel characteristics between coal and the co-firing material is an important factor in the co-firing process. The fuel ratio is typically calculated by the fixed carbon/volatile ratio; the accepted fuel ratio is generally 1.0–2.5 in thermoelectric power plants [44]. As shown in Table 1, a combination of FW and SS produces better biomass for co-firing than either FW or SS alone.

For the practical application of SRF in coal-fired plants, several issues should be addressed. First, this study reveals the potential of municipal solid waste-based biochar; however, the quality of fuel is not satisfactory and lacks various criteria. Further treatment, such as higher pyrolytic temperatures or different washing methods, should be employed to improve the performance of the fuel. Second, other properties of the fuel, such as moisture content and particle size, should be investigated. Moisture content is a critical aspect of fuel because it can reduce the calorific value and

lead to incomplete combustion [45,46]. Moreover, the broad particle size range of biochar may hinder the direct blending of biochar with coal. Potential agglomeration resulting from deficient blending could also increase $CO_2$ emissions [47].

## 4. Conclusions

This study explored the characteristics of biochar based on municipal organic waste (FW and SS) and its potential application as a co-firing material in a coal-fired plant. To optimize the biochar composition and pyrolysis conditions, different FW and SS mixing ratios and pyrolysis temperatures from 300–500 °C were tested. Both FW and SS showed clear advantages and disadvantages, which were reflected in the characteristics of the resulting biochar. A higher ratio of FW led to high chlorine contents and calorific value, whereas a higher ratio of SS led to increased heavy metal contents. Overall, all biochar samples exhibited successful fuel performance, especially with respect to calorific value. Pyrolysis and subsequent desalination improved the fuel properties of biochar by eliminating moisture and volatiles. However, chlorine and heavy metal concentrations should be further reduced to meet regulatory criteria, and a greater reduction of AAEMs concentrations is desirable for preventing slagging and fouling. To ensure efficient operation and similar or better energy output in thermoelectric plants, the co-pyrolysis of municipal solid waste and resulting fuel characteristics require further investigation.

**Author Contributions:** Conceptualization, methodology, and investigation, Y.J. and Y.-E.L.; data curation and writing—original draft preparation, Y.J.; writing—review and editing, I.-T.K., Y.J. and Y.-E.L.; supervision and funding acquisition, I-T.K. All authors have read and agreed to the published version of the manuscript.

**Funding:** This research was funded by the Korea Institute of Civil Engineering and Building Technology (KICT), grant number 20200166-001.

**Conflicts of Interest:** The authors declare no conflict of interest.

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
