# Peer review of "Characterization of Sewage Sludge and Food Waste-Based Biochar for Co-Firing in a Coal-Fired Power Plant: A Case Study in Korea"

_sustainability, doi:10.3390/su12229411_

Round 1

Reviewer 1 Report

This is a very well written paper. However, major concerns are expressed:

  • What is the novelty of this work? This is not obvious, since characterization of such a heterogeneous material such as food waste depends on the mixture of the food waste. What was the basis for selecting this mixture composition? I assume that the composition affects all characteristics, but this has not been taken into account… This means, what is the sensitivity analysis of varying the composition on the characteristics?
  • Aren’t there any studies of characterizing biochar in terms of calorific value, heavy metal, chlorine etc?
  • What was the origin of sewage sludge? Primary, secondary, mixture? Hasn’t the origin of sludge any impact on the characteristics?
  • There are no error bars nor standard deviations in the results. It seems that the tests were not repeated, and this is a major weakness.

Reviewer 2 Report

litearture review and motivation on the combined use of food waste (FW) and sewege sludge (SS)  to be provided.. what is  the source of sewage sludge (SS)? also, about is characteristics? biochar - do we use for energy generation?
What if it is used as negative emissions technologoies?
Selection of technlogies - using pyrolysis and desalination - Not clear! how do we combine food waste (FW)  and SS? Methods for analysis need to be improved. what is about uncertainity in the estimation? implication of derived results to be highlighted. Good luck for the revision!    

Reviewer 3 Report

Interesting research work with a high environmental and technical aplicability. The manuscript is well structured and the results are very interesting in order to valorisate sewage sludge and food waste. In addition, the data provided will be useful for other similar jobs, even for industrial production.

The number of references is adequate and both the state of the art and the discussion of results is well contrasted.

From my point of view, this article is a good fit for the journal "sustainability".

Author Response

Interesting research work with a high environmental and technical aplicability. The manuscript is well structured and the results are very interesting in order to valorisate sewage sludge and food waste. In addition, the data provided will be useful for other similar jobs, even for industrial production.

The number of references is adequate and both the state of the art and the discussion of results is well contrasted.

From my point of view, this article is a good fit for the journal "sustainability".

We thank the reviewer for positive feedback. We hope the revised version is better.

Reviewer 4 Report

I have any comments to presented results of the experiments concerning chemical and thermal properties of the tested sewage sludge and food waste samples and their mixtures.

The presented results concern only to the certain recommendations defined by Korean Institutions. I have only objection to giving the thermal units in kcal. In the scientific elaborations the SI units are commonly recommended in this case in KJ. So, I recommend to use everywhere KJ. 

Figure 3 shows molar distribution. Please give the units on Y-axis.

In Table 2 please give units (probably mass ratio).

Can you specify the contents of volatile components? The highest content of carbon 71,2% at 500 oC for FW=100% gives about 24.13 MJ/kg (5788 kcal/kg). But on the graph 1 low caloric value is 6500 kcal/kg. The rest should give hydrogen or sulfur. Please explain this fact.

I recommend also to show once again the experimental set-up, however, you have mentioned that was published earlier in [25]. But for potential readers showing this test stand is better solution. 

Author Response

I have any comments to presented results of the experiments concerning chemical and thermal properties of the tested sewage sludge and food waste samples and their mixtures.

The presented results concern only to the certain recommendations defined by Korean Institutions.

The results of this study were compared with both national and international standard of biomass solid refuse fuel (Bio-SRF). For practical operation of biochar, it is important to confirm that biochar satisfies national threshold of Bio-SRF. This is why national guideline has been extensively explained throughout the manuscript. However, the comparison with international standard (BS EN 15359:2011) has been also considered in this study (in L140-146, 158-159, 165-167). It seems the content regarding BS EN 15359:2011 is simplified because this guideline addresses only three factors: calorific value, chlorine content, and mercury content.

I have only objection to giving the thermal units in kcal. In the scientific elaborations the SI units are commonly recommended in this case in KJ. So, I recommend to use everywhere KJ.

The unit has been corrected. The unit MJ is used in the manuscript (Figure 1 and L112-113, L126-127, L141, and L143-144).

Figure 3 shows molar distribution. Please give the units on Y-axis.

Molar distribution in Figure 3 is unitless. This specific information has now been added in Figure 3.

In Table 2 please give units (probably mass ratio).

Units of all parameters have been now added in the first row of the Table 2.

Can you specify the contents of volatile components? The highest content of carbon 71,2% at 500 oC for FW=100% gives about 24.13 MJ/kg (5788 kcal/kg). But on the graph 1 low caloric value is 6500 kcal/kg. The rest should give hydrogen or sulfur. Please explain this fact.

The volatile matter is determined according to PN-EN 15148:2009. The sample is heated with ambient air at 900±10 ℃ for 7 min. The percentage of volatile matter is calculated based on the mass loss in tested sample after considering the mass loss of moisture. Accordingly, it is difficult to specify the contents of volatile components.

As mentioned by reviewer, FW100 sample pyrolyzed at 500℃ scored the highest fixed carbon content (71.2%). There are a number of models for estimating higher heating value (HHV) of biomass [1]. The predictable HHV ranged from 24.13 MJ/kg to 30.17 MJ/kg. As the calorific value is influenced by various factors including biomass type and physicochemical properties of biomass, the direct measurement of calorific value provides more accurate result than the estimation by models. Therefore, we believe that the measured calorific value of FW100 sample (27.13 MJ/kg) is acceptable and within the range of expectation.

I recommend also to show once again the experimental set-up, however, you have mentioned that was published earlier in [25]. But for potential readers showing this test stand is better solution.

We have included more details of experiment set-up (in lines 84-86 and L87-89).

[Reference]

  1. Vargas-Moreno, J.M.; Callejón-Ferre, A.J.; Pérez-Alonso, J.; Velázquez-Martí, B. A review of the mathematical models for predicting the heating value of biomass materials. Renew. Sustain. Energy Rev. 2012, 16, 3065–3083.

Round 2

Reviewer 1 Report

The authors have replied in detail but I am afraid the answers do not satisfy my major concerns.

1) Regarding the novelty of the study, my major concern is that the raw material for the biochar production is a mixture of mixtures. The mixture of FW to SS is well defined, but the individual mixtures are not homogeneous or standard. This is not so much problem for the SS, but it is for the FW. FW was based on a typical composition for the case of Korea which limits the merit of the study. FW composition is very much variable depending on several factors around the world. Moreover, the global efforts for minimising food waste, can alter the composition in the near future. From this aspect a sensitivity analysis is of major importance. The answer of the authors to this question " We confirmed that combination of food waste and sewage sludge offset the limitation of individual component and improve the performance of fuel". This confirmation is not presented in the manuscript or at least, it is not indicated in their reply where this is presented in the manuscript. 

2) Triplicate tests are missing. This means that all presented results are not certain since the standard deviation cannot be provided. Therefore, it is not certain if some points in the graphs that diverge from the apparent trend are "true" or lie within a large error interval. Standard deviation in this type of experiments are of outmost importance. 

Author Response

  • Regarding the novelty of the study, my major concern is that the raw material for the biochar production is a mixture of mixtures. The mixture of FW to SS is well defined, but the individual mixtures are not homogeneous or standard. This is not so much problem for the SS, but it is for the FW. FW was based on a typical composition for the case of Korea which limits the merit of the study. FW composition is very much variable depending on several factors around the world. Moreover, the global efforts for minimising food waste, can alter the composition in the near future. From this aspect a sensitivity analysis is of major importance. The answer of the authors to this question " We confirmed that combination of food waste and sewage sludge offset the limitation of individual component and improve the performance of fuel". This confirmation is not presented in the manuscript or at least, it is not indicated in their reply where this is presented in the manuscript.

Thank you for the comment. As the reviewer stated, the heterogeneous nature of food waste and sewage sludge has been one of critical issues in this field. In previous studies, to overcome this obstacle, the source of FW and SS is described in detail and physicochemical properties are provided to characterize the specific sample [1–5].

In this study, “Standard Food Waste Sample” was employed to represent the accustomed food waste in the Republic of Korea. We would like to point out the characteristic of food waste collection system in the Republic of Korea. Food waste is not collected together with other municipal wastes, but separately discharged and transported into the treatment facility dealing with food waste specifically.  As a large quantity of food waste (at least > 10 ton/day) is combined in the treatment facility, the heterogeneous nature of food waste from the origin can be offset and the composition of food waste is averaged. The ingredients and specific ratios of “Standard Food Waste Sample” are determined by ministry of environment [6] and these details are available in lines 75-76.

We also agree on the reviewer’s opinion that the composition of food waste can be altered in the future. We believe that further study in long-term basis will be required to sort out the feasible changes of food waste. In this respect, the presented results will be of interest to the researchers who explored the performance of biochar based on municipal solid wastes, the influence of pyrolysis temperature, and key aspects of cofiring material in thermoelectric power plant.

  • Triplicate tests are missing. This means that all presented results are not certain since the standard ddeviation cannot be provided. Therefore, it is not certain if some points in the graphs that diverge from the apparent trend are "true" or lie within a large error interval. Standard deviation in this type of experiments are of outmost importance.

We acknowledge that this is main limitation of this study. Although triplicate analysis was not available due to limited volume of the sample, we consider that the presented results are still valid and worth to be published. First, the determination of individual experimental value is accurate and robust. Triplicate measurements of ICP-MS and ICP-AES were conducted and the robustness of analysis was confirmed by relative standard deviation (RSD, %). As an example, the RSD data of Na, K, Ca, and Mg is presented in Figure 1, all of which is less than 4%. Second, the trends of experiment result with pyrolysis temperature and the mixed ratio between FW and SS are consistent in this study. For example, increasing SS ratio leads to lower calorific value and higher heavy metal content. These results are in accord with those of previous studies. The discussions on this agreement are in the manuscript (L128-132 and L245-246).

[Reference]

  1. Wu, S.; Xu, S.; Chen, X.; Sun, H.; Hu, M.; Bai, Z.; Zhuang, G.; Zhuang, X. Bacterial Communities Changes during Food Waste Spoilage. Sci. Rep. 2018, 8, 8220.
  2. Du, C.; Abdullah, J.J.; Greetham, D.; Fu, D.; Yu, M.; Ren, L.; Li, S.; Lu, D. Valorization of food waste into biofertiliser and its field application. J. Clean. Prod. 2018, 187, 273–284.
  3. Fu, M.-M.; Mo, C.-H.; Li, H.; Zhang, Y.-N.; Huang, W.-X.; Wong, M.H. Comparison of physicochemical properties of biochars and hydrochars produced from food wastes. J. Clean. Prod. 2019, 236, 117637.
  4. Borowski, S.; Boniecki, P.; Kubacki, P.; Czyżowska, A. Food waste co-digestion with slaughterhouse waste and sewage sludge: Digestate conditioning and supernatant quality. Waste Manag. 2018, 74, 158–167.
  5. Wickham, R.; Xie, S.; Galway, B.; Bustamante, H.; Nghiem, L.D. Anaerobic digestion of soft drink beverage waste and sewage sludge. Bioresour. Technol. 2018, 262, 141–147.
  6. Ministry of Environment A Study on food waste reduction equipment guidelines and quality standard; Sejong City, 2009;

Round 3

Reviewer 1 Report

I am afraid that the major weakness of the missing triplicates still remains. The measurements may be valid, since there are triplicates verifying the accuracy of the measurements. However, the repeatability of the tests is questionable, especially due to the high heterogeneity of the MSW. Such tests would be meaningful, if samples were taken even in different days, when composition may be a little different. But even from the same sample, repitition of the tests are necessary to check of the final mixtures and the results of the processing are the same, or different and to what extent. This is a typical "must" in these kind of tests.

Concerning the other weakness which is the local and temporal character of the research (since MSW composition is typical locally and for the time being), this could be considered a minor, but the title should reflect that this is a case study.

Author Response

We appreciate the reviewer for thoroughly reading our manuscript and providing helpful feedbacks. All comments have been considered and incorporated in the revised manuscript. The changes and additions are highlighted in yellow and the response to reviewer’s comments are written in red.

The heterogeneous nature of municipal solid wastes is indeed critical issue. To overcome this, detailed explanation of experimental procedures and samples has been added in lines 75-83. We have now modified title “Characterization of sewage sludge and food waste-based biochar for co-firing in a coal-fired power plant: a case study in Korea”. In further study, multiple measurements are planned to provide reliable physicochemical properties of biochar.  
